# Relationship between Chewing Status and Fatty Liver Diagnosed by Liver/Spleen Attenuation Ratio: A Cross-Sectional Study

**DOI:** 10.3390/ijerph20010307

**Published:** 2022-12-25

**Authors:** Komei Iwai, Tetsuji Azuma, Takatoshi Yonenaga, Yasuyuki Sasai, Kazutoshi Watanabe, Fumiko Deguchi, Akihiro Obora, Takao Kojima, Takaaki Tomofuji

**Affiliations:** 1Department of Community Oral Health, School of Dentistry, Asahi University, Mizuho 501-0296, Japan; 2Asahi University Hospital, Gifu 500-8523, Japan

**Keywords:** fatty liver, mastication, eating behavior, X-ray computed tomography, cross-sectional study

## Abstract

This cross-sectional study investigated the relationship between chewing status and fatty liver among Japanese adults. Between April 2018 and March 2021, 450 individuals (352 males, 98 females; mean age 54.7 years) were recruited at the Asahi University Hospital Human Health Center. Chewing status was evaluated using a self-reported questionnaire. Liver/spleen (L/S) attenuation ratio < 0.9 on computed tomography was considered to indicate fatty liver, which was present in 69 participants (15%). Compared with participants without fatty liver, those with fatty liver had higher proportion of 25.0 (kg/m^2^) ≤ body mass index (BMI) (*p* < 0.001), higher serum hemoglobin A1c (HbA1c) (*p* < 0.001), higher systolic blood pressure (*p* < 0.001), higher diastolic blood pressure (*p* < 0.001), and lower serum high-density lipoprotein cholesterol (HDL cholesterol) (*p* = 0.011). Significant differences were also found in chewing status (*p* < 0.001) and eating speed (*p* = 0.011). Presence of fatty liver was positively associated with BMI (25.0 ≤; odds ratio [OR], 5.048; 95% confidence interval [CI], 2.550–9.992), serum HbA1c (OR, 1.937; 95% CI, 1.280–2.930), and chewing status (poor; OR, 8.912; 95% CI, 4.421–17.966) after adjusting for sex, age, BMI, serum HbA1c, systolic blood pressure, diastolic blood pressure, serum HDL cholesterol, chewing status, and eating speed. Poor chewing status was positively associated with L/S attenuation ratio. These results indicate a positive relationship between poor chewing status and fatty liver diagnosed by L/S attenuation ratio in Japanese adults.

## 1. Introduction

Fatty liver is the condition of abnormal accumulation of fat in the liver [1]. It is caused by abnormalities in lipid metabolism, including increased uptake of lipids into the liver, decreased lipolysis, and decreased fat delivery [2]. In its early stages, fatty liver produces few subjective symptoms such as pain [3]. However, as symptoms progress, it can lead to liver failure, kidney failure, and liver cancer [4,5,6]. Understanding the risk factors for fatty liver and preventing its development is therefore crucial for the avoidance of these serious diseases.

It was reported that fatty liver is caused by overweight, obesity, alcohol consumption, and excessive dieting [7,8]. Furthermore, previous studies have revealed a relationship between eating behavior and hepatic health. Two cross-sectional studies found a significant association between eating speed and nonalcoholic fatty liver disease [9,10], whereas another reported that subjects who habitually ate before bedtime, and those who ate quickly and before bedtime, tended to have an increased risk of nonalcoholic fatty liver disease [11]. These findings indicate that eating behavior is potentially related to fatty liver.

Chewing status is another health condition related to eating behavior. Previous studies have shown an association of chewing status with obesity and glycemic control [12,13]. Similarly, chewing status may also be associated with fatty liver. However, there is little information available regarding the relationship between chewing status and fatty liver, and further clinical studies are necessary.

We hypothesized that chewing status might be associated with the presence of fatty liver. In Japan, the Ministry of Health, Labour and Welfare requires medical insurers to provide specific health checkups focusing on lifestyle habits to insured persons aged 40–74 years [14,15]. The questionnaire in specific health checkups was developed based on the conventional National Health and Nutrition Examination Survey and questions specified by the Industrial Safety and Health Act. It includes items on eating behavior, including chewing status. Liver/spleen (L/S) attenuation ratio by computed tomography (CT) is an established method for detecting the presence of fatty liver [16,17,18]. At the Asahi University Hospital Human Health Center, it is possible to collate data from specific health checkups and from abdominal CT scans. Therefore, the purpose of this study was to investigate the relationship between self-reported chewing status and fatty liver diagnosed by L/S attenuation ratio in Japanese adults who underwent medical health checkup at the Asahi University Hospital Human Health Center.

## 2. Materials and Methods

### 2.1. Participants

We recruited 454 individuals who underwent specific medical checkups and abdominal CT scans at the Asahi University Hospital Human Health Center between April 2018 and March 2021. Of these, 4 participants were excluded because of missing CT value data for the spleen. A final total of 450 participants (352 males and 98 females, mean age 54.7 years) were included in the analysis.

### 2.2. Evaluation of Chewing Status and Other Items

We used the same questionnaire used in the specific medical checkup in Japan [19]. The questionnaire items included chewing status (“I can eat anything”, “Sometimes it is difficult to chew because of dental problems such as dental caries and periodontal disease”, or “I can hardly chew”). In categorizing chewing status, respondents who answered “I can eat anything” were categorized as “good”; those who answered “Sometimes it is difficult to chew because of dental problems such as dental caries and periodontal disease” as “sometimes difficult”; and those who answered “I can hardly chew” as “difficult” [12]. Other items collected from the specific medical checkup questionnaire were sex, age, smoking habit (smoking at least one cigarette per day: presence or absence), alcohol consumption (current alcohol consumption: “<2 go [180 mL/go]” or “≥2 go” per day, where “go” is a traditional Japanese unit of volume measurement, corresponding to 23 g of ethanol), exercise habit (“I have engaged in ≥30 min of light exercise more than twice/week for ≥1 year”: presence or absence), physical activity (“I go for a walk or perform an equivalent physical activity for ≥1 h/day”: presence or absence), sleep status (poor or good), eating speed (slow, medium, or fast), snacking habit (none, sometimes, or daily), breakfast skipping habit (<3 times/week or ≥3 times/week), and habit of having dinner within 2 h before bedtime (<3 times/week or ≥3 times/week) [19,20,21,22]. Alcohol consumption of ≥2 go per day was defined as “heavy alcohol consumption” [21].

### 2.3. Evaluation of the Presence of Fatty Liver

All participants underwent plain abdominal CT (FUJIFILM, Tokyo, Japan) after overnight fasting. A region of interest was set at three positions in each of the liver and spleen, avoiding blood vessels and heterogeneous areas, and attenuation (in Hounsfield units) was measured at each point. The mean attenuation value of each of the liver and spleen was used to calculate the L/S attenuation ratio. The criterion for diagnosis of fatty liver was L/S attenuation ratio < 0.9 [16,17,18].

### 2.4. Assessment of Body Composition

During the specific health checkups, nurses measured each participant’s height and body weight. Body mass index (BMI) was calculated as weight divided by the square of height (kg/m^2^). In Japan, people with BMI of 25.0 (kg/m^2^) or more are defined as obese [23]. Therefore, participants were classified into two groups: those with <25.0 or 25.0 ≤ BMI.

### 2.5. Measurement of Blood Pressure

Nurses measured systolic and diastolic blood pressure values of each participant. Measurements were made according to the guidelines of the Japanese Society of Hypertension, as follows [24,25]. Measurements were taken with a brachial automatic sphygmomanometer (TM-2657VP; A & D Co., Tokyo, Japan) by placing the forearm on a support stand, wrapping the lower end of the cuff 2–3 cm above the elbow, and aligning the center of the cuff with the level of the heart. Furthermore, measurements were taken twice with an interval of 1–2 min. If the two measurements differed significantly (more than 5 mmHg), measurements were taken again. The average of the two stable values was used as the blood pressure value.

### 2.6. Other Measurements

Serum HbA1c level, serum triglyceride level, and serum HDL cholesterol level were measured by high-performance liquid chromatography in venous blood samples collected after an overnight fast [26,27].

### 2.7. Statistical Analysis

Continuous variables (age, serum HbA1c level, systolic blood pressure level, diastolic blood pressure level, serum triglyceride level, serum HDL cholesterol level, and L/S attenuation ratio) are expressed as the median (25%, 75% percentiles) because these variables are not normally distributed. The relationship between each variable and the presence or absence of fatty liver was assessed using Fisher’s exact test and the Mann–Whitney U-test. Univariate and multivariate logistic regression analyses were performed with presence of fatty liver as the dependent variable. Among the participants with fatty liver, few (*n* = 12) answered “I can hardly chew”. Thus, we combined “sometimes difficult” and “difficult” into the “poor” chewing status category. For similar reasons, eating speed was dichotomized as “fast” or “not fast” and snacking habit as “daily” or “not daily”. Chewing status was set as the independent variable in univariate and multivariate logistic regression analyses as “good” or “poor”. Variables with *p* < 0.10 were excluded from the model. In addition to variables related to the sample (age, sex), variables with significant differences in univariate logistic analysis (BMI, serum HbA1c level, systolic blood pressure level, diastolic blood pressure level, serum HDL cholesterol level, chewing status, and eating speed) were set as confounding factors, which were adjusted for in these analyses. Multivariate logistic regression analysis used the variable backward method. The relationship between L/S attenuation ratio and chewing status was assessed using the Mann–Whitney U-test. All data were analyzed using a statistical analysis software (SPSS statistics version 27; IBM Japan, Tokyo, Japan). All *p*-values < 0.05 were considered statistically significant.

### 2.8. Research Ethics

Our study was approved by the Ethics Committee of Asahi University (No. 27010), and was performed in accordance with the Declaration of Helsinki. Written informed consent was obtained from all participants.

## 3. Results

Table 1 shows the characteristics of all participants, with and without fatty liver. Of the 450 participants, 69 (15%) had fatty liver and 61 (13.5%) described their chewing status as “poor”. Compared with participants without fatty liver, those with fatty liver were more likely to have higher proportion of 25.0 ≤ BMI (*p* < 0.001), higher serum HbA1c level (*p* < 0.001), higher systolic blood pressure level (*p* < 0.001), and higher diastolic blood pressure level (*p* < 0.001). In addition, participants with fatty liver were more likely than those without fatty liver to have lower serum HDL cholesterol level (*p* = 0.011), and there were significant differences among the two groups in terms of chewing status (*p* < 0.001) and eating speed (*p* = 0.011).

The results of logistic regression analysis with fatty liver as the dependent variable are shown in Table 2. In the crude model, the presence of fatty liver was significantly associated with BMI (25.0 ≤; OR, 6.612; 95% CI, 3.751–11.659), serum HbA1c level (higher; OR, 2.104; 95% CI, 1.454–3.045), systolic blood pressure level (higher; OR, 1.033; 95% CI, 1.015–1.052), diastolic blood pressure level (higher; OR, 1.004; 95% CI, 1.019–1.071), serum HDL cholesterol level (higher; OR, 0.979; 95% CI, 0.963–0.995), chewing status (poor; OR, 7.202; 95% CI, 3.958–13.104), and eating speed (fast; OR, 1.785; 95% CI, 1.036–3.075). In the adjusted model, the presence of fatty liver was significantly associated with BMI (25.0 ≤; OR, 5.048; 95% CI, 2.550–9.992), serum HbA1c level (higher; OR, 1.937; 95% CI, 1.280–2.930), and chewing status (poor; OR, 8.912; 95% CI, 4.421–17.966) after adjusting for sex, age, BMI, serum HbA1c level, systolic blood pressure level, diastolic blood pressure level, serum HDL cholesterol level, chewing status, and eating speed.

Figure 1 shows the L/S attenuation ratio values for good and poor chewing status. Compared with participants with good chewing status, those with poor chewing status (*n* = 61, 14%) were more likely to have a low L/S attenuation ratio (*p* < 0.001).

## 4. Discussion

To the best of our knowledge, the present study is the first to examine the association between self-reported chewing status and fatty liver diagnosed by L/S attenuation ratio in Japanese adults. The results showed that compared with those without fatty liver, a higher proportion of participants with fatty liver had poor chewing status.

Our analyses also revealed that presence of fatty liver was associated with poor chewing status after adjusting for sex, age, BMI, serum HbA1c level, systolic blood pressure level, diastolic blood pressure level, serum HDL cholesterol level, and eating speed. Furthermore, participants with poor chewing status had a lower L/S attenuation ratio than those without poor chewing status. These observations suggest that people with self-reported poor chewing status are more likely to have fatty liver. It is feasible that poor chewing status may be a risk factor for fatty liver in Japanese adults.

The gold standard diagnostic method for fatty liver is liver biopsy, but this method carries risks. Imaging methods offer non-invasive alternatives to liver biopsy, among which the L/S attenuation ratio on CT has been shown to have accuracy comparable to that of tissue diagnosis [28]. A CT L/S attenuation ratio cut-off value of 0.9 has been reported to have sensitivity of 79% and specificity of 97% for detection of severe fatty liver [29]. Therefore, in the present study, a L/S attenuation ratio of <0.9 was considered the cutoff value for diagnosing the presence or absence of fatty liver [16,17,18].

There are some possible mechanisms for the relationship between chewing status and fatty liver. Individuals with poor chewing status tend to consume less fruit and vegetables and more high-energy foods than those with good chewing status [30], which can lead to fatty liver. In addition, chewing increases blood level of glucagon-like peptide-1 levels and promotes insulin secretion [31]. Since the function of liver changes according to the blood concentration of insulin [32], people with poor chewing status may induce fat accumulation in the liver via reducing insulin secretion. Furthermore, chewing has the function of activating the histaminergic nervous system in the brain through mechanoreceptive sensation from periodontal ligament, suppressing appetite and promoting visceral lipolysis or promoting body heat production [33,34]. Thus, participants with poor chewing status may be suppressed in these effects, which may increase the amount of fat in the body and make them more susceptible to fatty liver.

Previous studies have reported a relationship between chewing status and systemic health. For instance, a systematic review reported that in 12 of 16 cross-sectional studies, poorer chewing status was associated with obesity [35]. It is also known that poor chewing status is associated with poor glycemic control [12]. The previous and present studies support the notion that poor chewing status could be detrimental to systemic health.

In our study, the presence of fatty liver was positively associated with higher BMI and higher serum HbA1c level. These observations are consistent with previous studies that reported an association between fatty liver and higher BMI [36,37] and an association between fatty liver and higher HbA1c [38].

The proportion of the present participants with fatty liver was 15%. In Japan, the prevalence of fatty liver is reported to range from 9% to 30% [39,40]. Therefore, the prevalence in our study may represent an average group in Japan. However, the external validity of our study should be considered, because all participants were recruited from the Asahi University Hospital Human Health Center.

Our study has some further limitations. First, as our study was cross-sectional in nature, we could not demonstrate causal relations. Additional longitudinal studies are needed to investigate the relationship between chewing status and fatty liver. Second, we evaluated chewing status based on a questionnaire. Therefore, there may be a discrepancy between self-reported chewing status and actual chewing problems. However, it has also been reported that chewing status confirmed by a self-administered questionnaire is not only related to number of present, molar, and functional teeth, but is also useful as a screening method for actual chewing ability [41]. Third, it is difficult to figure out what causes poor chewing status for teeth. Fourth, the diagnosis of alcoholic fatty liver requires the percentage of triglycerides in liver cells and the duration of alcohol consumption [42], but these data are not available because they are not part of specific health examination items. Finally, dietary factors or calorie intake of participants in our study is unknown.

## 5. Conclusions

Self-reported poor chewing status was associated with the presence of fatty liver diagnosed by L/S attenuation ratio in Japanese adults. Checking self-reported chewing status may be useful in screening for the early detection of fatty liver.

## Figures and Tables

**Figure 1 ijerph-20-00307-f001:**
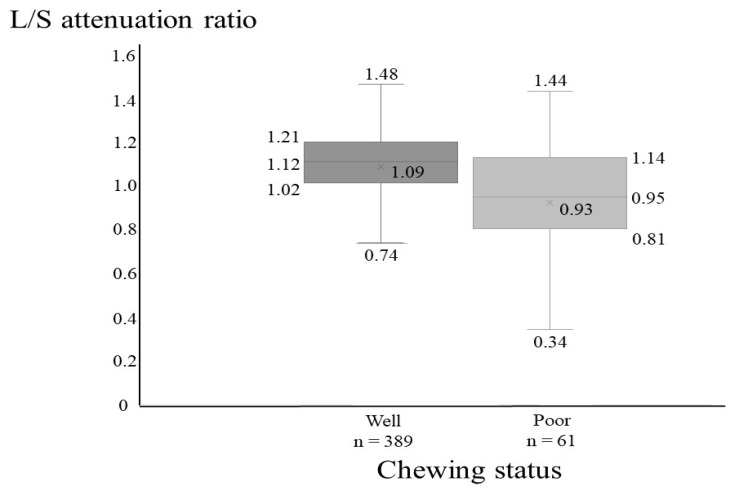
Relationship between L/S attenuation ratio and chewing status. Abbreviations: L/S attenuation ratio, liver/spleen attenuation ratio. *p* < 0.001, Mann–Whitney *U*-test.

**Table 1 ijerph-20-00307-t001:** Participant characteristics according to absence or presence of fatty liver.

Factor	Fatty Liver		*p*-Value *
	Absence	Presence	
	(*n* = 381)	(*n* = 69)	
Male ^a^	293 (77%)	59 (86%)	0.111
Age (years)	55 (45, 62)	56 (45, 59)	0.603
BMI (kg/m^2^)			
<25.0	278 (73%)	20 (29%)	<0.001
25.0 ≤	103 (27%)	49 (71%)	
Smoking habits ^b^	50 (13%)	13 (19%)	0.208
Alcohol consumption ^c^	80 (21%)	15 (22%)	0.89
Exercise habit ^b^	208 (55%)	40 (58%)	0.604
Physical activity ^b^	162 (43%)	29 (42%)	0.94
Serum HbA1c level (%)	5.5 (5.3, 5.7)	5.6 (5.3, 6.1)	<0.001
Systolic blood pressure level (mmHg)	122 (112, 131)	130 (122, 137)	<0.001
Diastolic blood pressure level (mmHg)	76 (68, 84)	83 (74, 88)	<0.001
Serum triglyceride level (mg/dL)	74 (51, 116)	96 (67, 136)	0.158
Serum HDL cholesterol level (mg/dL)	60 (50, 74)	56 (48, 66)	0.011
Sleep status ^d^	83 (22%)	16 (23%)	0.796
Chewing status			
Good	348 (91%)	41 (59%)	<0.001
Sometimes Difficult	30 (8%)	19 (28%)	
Difficult	3 (1%)	9 (13%)	
Eating speed			
Slow	42 (11%)	1 (1%)	0.011
Medium	247 (65%)	43 (62%)	
Fast	92 (24%)	25 (37%)	
Snacking habit			
None	49 (13%)	9 (13%)	0.985
Sometimes	268 (70%)	49 (71%)	
Daily	64 (17%)	11 (16%)	
Skipping breakfast habit			
<3 times/week	360 (95%)	62 (90%)	0.143
≥3 times/week	21 (5%)	7 (10%)	
Dinner at least 2 h before bedtime			
<3 times/week	286 (75%)	46 (67%)	0.144
≥3 times/week	95 (25%)	23 (33%)	

Abbreviations: BMI, body mass index; serum HbA1c level, serum hemoglobin A1c level; serum HDL cholesterol level, serum high density lipoprotein cholesterol level. * *p* < 0.05, using Fishers exact test or Mann–Whitney U-test. ^a^ Male (proportion of male); ^b^ Presence (proportion of presence); ^c^ Heavy (proportion of heavy); ^d^ Poor (proportion of poor).

**Table 2 ijerph-20-00307-t002:** Crude and adjusted odds ratios and 95% CI for fatty liver.

Factor		ORs	95% Cl	*p*-Value
**Crude Model**				
Sex	Female	1	(reference)	0.115
	Male	1.772	0.870–3.609	
Age (years)		0.994	0.970–1.018	0.603
BMI (kg/m^2^)	<25.0	1	(reference)	<0.001
	25.0 ≤	6.612	3.751–11.659	
Smoking habits	Absence	1	(reference)	0.211
	Presence	1.537	0.784–3.011	
Alcohol consumption	Not heavy	1	(reference)	0.89
	Heavy	1.045	0.561–1.948	
Exercise habit	Absence	1	(reference)	0.604
	Presence	1.147	0.683–1.927	
Physical activity	Absence	1	(reference)	0.94
	Presence	0.98	0.583–1.648	
Serum HbA1c level (%)		2.104	1.454–3.045	<0.001
Systolic blood pressure level (mmHg)		1.033	1.015–1.052	<0.001
Diastolic blood pressure level (mmHg)		1.004	1.019–1.071	<0.001
Serum triglyceride level (mg/dL)		1.002	0.999–1.005	0.158
Serum HDL cholesterol level (mg/dL)		0.979	0.963–0.995	0.011
Sleep status	Good	1	(reference)	0.796
	Poor	1.084	0.589–1.994	
Chewing status	Good	1	(reference)	<0.001
	Poor	7.202	3.958–13.104	
Eating speed	Not fast	1	(reference)	0.037
	Fast	1.785	1.036–3.075	
Snacking habit	Not daily	1	(reference)	0.741
	Daily	0.885	0.429–1.825	
Skipping breakfast habit	<3 times/week	1	(reference)	0.149
	≥3 times/week	1.935	0.789–4.746	
Dinner at least 2 h before bedtime	<3 times/week	1	(reference)	0.146
	≥3 times/week	1.505	0.867–2.614	
**Adjusted Model**				
Sex	Female	1	(reference)	0.567
	Male	0.762	0.300–1.934	
Age (years)		0.985	0.954–1.018	0.37
BMI (kg/m^2^)	<25.0	1	(reference)	<0.001
	25.0 ≤	5.048	2.550–9.992	
Serum HbA1c level (%)		1.944	1.286–2.937	0.002
Systolic blood pressure level (mmHg)		1.02	0.966–1.076	0.476
Diastolic blood pressure level (mmHg)		1.013	0.973–1.054	0.526
Serum HDL cholesterol level (mg/dL)		0.995	0.972–1.017	0.63
Chewing status	Good	1	(reference)	<0.001
	Poor	8.912	4.421–17.966	
Eating speed	Not fast	1	(reference)	0.58
	Fast	1.231	0.649–2.336	

Abbreviations: ORs, odds ratios; CI, confidence interval; BMI, body mass index; serum HbA1c level, serum hemoglobin A1c level; serum HDL cholesterol level, serum high density lipoprotein cholesterol level. Adjusted Model; Adjustment for sex, age, BMI, HbA1c level, systolic blood pressure level, diastolic blood pressure level, HDL cholesterol level, chewing status, and eating speed.

## Data Availability

The data are not publicly available due to restrictions for reasons of privacy and ethics. The data presented in this study are available on request from the corresponding author.

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
