# Peer review of "Relationship between Chewing Status and Fatty Liver Diagnosed by Liver/Spleen Attenuation Ratio: A Cross-Sectional Study"

_ijerph, 2022, doi:10.3390/ijerph20010307_

Round 1

Reviewer 1 Report

Thank you for your interesting manuscript. However, I have some comments for it.

1.      Concerning with methodology and results, the authors should clearly present the causes of fatty liver in the introduction. Then, you could follow this frame in the presentation of results.  Generally, the causes of fatty livers are around four. E.g the overweight and obesity are two of them, so you should have categorized the BMI into obesity category according to WHO or somewhat you want to refer and reported it.

2.      Systolic and diastolic blood pressure are also key players for fatty liver. Thus, the precise measurement in the methodology are important. You should add more information about it although you had cited the reference.

3.      It is better to include your results briefly presented in fatty liver, that focused on alcoholic or non-alcoholic.

4.      Regarding non-alcoholic fatty liver, excess calorie intake is one main cause. However, that point is missing in your study. In my point of view, it is difficult to say any association between chewing status and fatty liver while missing calorie intake.

5.      In the table 3, you presented the adjusted OR. How is the method for logistic regression, I mean which method you used, (E.g DAG/ Forward / Backward)? You should reveal it in the methodology. Which variables are adjusted for each independent variable? How did you manage the bias, confounding of the study?

Author Response

1. Concerning with methodology and results, the authors should clearly present the causes of fatty liver in the introduction. Then, you could follow this frame in the presentation of results. Generally, the causes of fatty livers are around four. E.g the overweight and obesity are two of them, so you should have categorized the BMI into obesity category according to WHO or somewhat you want to refer and reported it.

Response: We thank the reviewer for your valuable advice. We have revised our methodology and results according to your suggestions. We have added references and included them in introductions (lines 37-38, References 7, 8).

In Japan, people with BMI of 25.0 (kg/m2) or more is defined as obesity. Therefore, we have also reanalyzed by classifying participants into two groups: those with < 25.0 or 25.0 BMI. We have revised the sentences and table (lines 98-99, 146, 154, 155, 160, Table1, 2, References 23).

2. Systolic and diastolic blood pressure are also key players for fatty liver. Thus, the precise measurement in the methodology are important. You should add more information about it although you had cited the reference.

Response: We thank the reviewers for your valuable advice. We have followed your advice and added text and references regarding the accurate measurement of blood pressure (lines 102-109, References 24, 25).

3. It is better to include your results briefly presented in fatty liver, that focused on alcoholic or non-alcoholic.

Response: We thank the reviewer for your valuable advice. The diagnosis of alcoholic fatty liver requires the percentage of triglycerides in liver cells and the duration of alcohol consumption, but these data are not available because they are not part of specific health examination items. Therefore, proportion of alcoholic and non-alcoholic fatty liver in the present participants is unknown. We have included that sentence in limitations (lines 226-228).

4. Regarding non-alcoholic fatty liver, excess calorie intake is one main cause. However, that point is missing in your study. In my point of view, it is difficult to say any association between chewing status and fatty liver while missing calorie intake.

Response: We thank the reviewer for your valuable advice. As you pointed out, excess calorie intake is one main cause of fatty liver. This is a limitation of our study (lines 228-229). However, people who intake excess calorie usually become obesity. In Japan, people with 25.0 ≤ BMI are diagnosed as being obesity, and lifestyle guidance such as calorie restriction is provided. In this study, the results showed that poor chewing status and fatty liver were independently associated after adjusting presence or absence of 25.0 ≤ BMI (lines146, 154-155, 159-163, Table1, 2).

5. In the table 3, you presented the adjusted OR. How is the method for logistic regression, I mean which method you used, (E.g., DAG/ Forward / Backward)? You should reveal it in the methodology. Which variables are adjusted for each independent variable? How did you manage the bias, confounding of the study?

Response: We thank the reviewer for your valuable advice. In this study, multivariate logistic regression analysis used the variable backward method. We have added description of the adjusted variables and confounding factors in methodology (lines 128-132).

Reviewer 2 Report

This research topic is about chewing discomfort and fatty liver, which is a creative research topic.

However, it is necessary to supplement the consideration by referring to previous studies.

1) Evaluation of chewing status and other items - Are there standard or criterion for exercise habits and physical activities?

Please provide a reference for that criterion.

2) If possible, I recommend a table that combines Table 2 and Table 3.

3) discussion

These data were performed on patients who visited the hospital.

Therefore, I think it is unreasonable to compare with the National Health and Nutrition Examination Survey.

(This is because it did not consider the selection of subjects or the sample.)

4) Discussion

Please add limitations of this study.

1) It is difficult to figure out what causes chewing discomfort for teeth

2) Dietary factors were not considered.

5) It is necessary to supplement the effect that chewing discomfort can have on fatty liver.

Supplement with previous research.

Author Response

Evaluation of chewing status and other items - Are there standard or criterion for exercise habits and physical activities? Please provide a reference for that criterion.

Response: We thank the reviewer for your valuable advice. Yes, there are criterions for exercise habits and physical activities (lines 80-82). We have cited the reference (References 19-22).

If possible, I recommend a table that combines Table 2 and Table 3.

Response: We thank the reviewer for your valuable advice. In accordance with your advice, we have combined Tables 2 and 3 (new Table 2). We have also revised the sentences (lines154-163).

Discussion: These data were performed on patients who visited the hospital. Therefore, I think it is unreasonable to compare with the National Health and Nutrition Examination Survey. (This is because it did not consider the selection of subjects or the sample.)

Response: We thank the reviewer for your valuable advice. We have removed the sentences to avoid misleading.

Discussion: Please add limitations of this study.

1) It is difficult to figure out what causes chewing discomfort for teeth.

2) Dietary factors were not considered.

Response: We thank the reviewer for your valuable advice. In accordance with your advice, we have included your sentence in limitations (lines 225-226, 228-229).

It is necessary to supplement the effect that chewing discomfort can have on fatty liver. Supplement with previous research.

Response: We thank the reviewer for your valuable advice. In accordance with your suggestion, we have added the words “In addition, chewing increases blood level of glucagon-like peptide-1 levels and promotes insulin secretion. Since the function of liver changes according to the blood concentration of insulin, people with poor chewing status may induce fat accumulation in the liver via reducing insulin secretion. Furthermore, chewing has function of activating histaminergic nervous system in brain through mechanoreceptive sensation from periodontal ligament, suppressing appetite and promoting visceral lipolysis or promoting body heat production. Thus, participants with poor chewing status may be suppressed in these effects, which may increase the amount of fat in body and make them more susceptible to fatty liver” (lines 196-204).